# Comparative Cutting Fluid Study on Optimum Grinding Parameters of Ti-6Al-4V Alloy Using Flood, Minimum Quantity Lubrication (MQL), and Nanofluid MQL (NMQL)

Jose Jaime Taha-Tijerina *[ID] and Immanuel A. Edinbarough

Department of Informatics and Engineering Systems, The University of Texas Rio Grande Valley, Brownsville, TX 78520, USA; immanuel.edinbarough@utrgv.edu
* Correspondence: jose.taha@utrgv.edu

**Abstract:** Titanium alloys have been of paramount interest to the aerospace industry due to their attractive characteristics. However, these alloys are difficult to machine and require grinding post-processes for quality assurance of the products. Conventional grinding takes a long time and uses a flood coolant-lubrication technique, which is not cost effective nor environmentally friendly. Several studies have been performed to prove the viability and benefit of using Minimum Quantity Lubrication (MQL) with vegetable or synthetic-ester fluids. This work aims to find the optimum grinding parameters of creep feed grinding Ti-6Al-4V with a green silicon carbide wheel, using a flood lubrication system with water-soluble synthetic oil, MQL with ester oil, and nano-MQL (NMQL) using alumina-nanopowder homogeneously dispersed within an ester oil. It is concluded that at 0.635 mm and 1.27 mm infeeds, the three lubrication methods performed similarly. At an infeed of 1.905 mm, MQL did not provide desirable quality, though NMQL and flood lubrication performed practically identically. At a cross feed of 0.254 mm, an infeed of 1.27 mm, and a table feed rate of 6.7 m/min, these grinding parameters provide a material removal rate of 2163 mm$^3$/min with a surface roughness across (Ra) of 0.515 μm. These parameters provide the quickest material removal rate while still maintaining industrial quality. This conclusion is based on environmental, economic, and qualitative results.

**Keywords:** grinding; MQL; nano cutting fluid; Ti-6Al-4V titanium alloy; surface quality

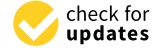



## 1. Introduction

Nanotechnology has quickly evolved, and developments in nanomaterials have been applied in machining and grinding manufacturing operations to improve manufacturing conditions as well as tooling performance and working pieces quality. Solid nanostructures can be incorporated into the working fluid to form novel grinding reinforced fluid with improved properties and characteristics. Homogeneous dispersion of nanostructures within fluids and lubricants could change the heat transfer attributes of the working fluid and lubricants, mechanical performance of the metal-metal interface, and path of the heat dissipation, thus modifying the temperature and energy efficiency of the tooling and workpiece surface [1–6].

Ti-6Al-4V is a titanium alloy known for its high strength, low density, non-toxicity, and anti-corrosive properties, and is a preferred metal in many industries from space, military, aviation, automotive, marine, and biomedical. Many industries are beginning to use Ti-6Al-4V as their material of choice in manufacturing due to these desirable properties. However, titanium manufacturing—machining costs are inhibiting further growth in these markets. Titanium is complex to process, machining, due to its high chemical affinity, which makes the chips and debris easily adhere onto the tooling due to very high localized temperature and extreme pressure. This behavior affects the service life of tooling and produces low surface roughness [7–9]. An alternative to improve

the fluid's cooling performance is to enhance the heat transfer surface area, as well as increasing the velocity of the fluid (high pressure). Nevertheless, this approach requires more energy consumption of pumps and substantial volume usage of fluid, which is not cost effective [10,11].

Developments in nanotechnology have led to the advent of fluids and lubricants with high thermal conductivity. Nanofluids and nanolubricants are engineered colloidal suspensions of nanostructures homogeneously dispersed within conventional fluids. The application of the nanofluids as coolants has shown significant improvements in the thermo-physical characteristics, which is mainly due to the inclusion of reinforcing nanostructures and their molecular interaction among the base fluids and lubricants [12–15]. Nanofluids have attracted the interest of scientists and industry due to their superb thermal transport characteristics. For instance, development of novel nanofluid by Taha-Tijerina et al. [16], where adding merely 0.10 wt.% of exfoliated hexagonal boron nitride (h-BN) nanosheets within mineral oil (MO) showed superb improvement in thermal conductivity of ~80%, as compared to pure MO. In similar studies by Mehrali et al. [17], good effects on the thermal conductivity of graphene-reinforced nanofluids were observed. A 45% enhancement at 4.0 vol.% was shown when compared to conventional red wine polyphenol solution. Commonly, to achieve such enhancements, conventional particle–liquid suspensions require high particle concentrations (>10%). However, as a higher reinforcing particle concentration is present, several drawbacks and issues could affect the fluids and lubricants acting on diverse systems and devices. These problems affect rheology and stability, in which these issues could be amplified and particles would tend to agglomerate and sediment faster, lowering thermal transport performance and precluding the application of conventional materials for heat transfer.

Grinding is an abrasive material removal process, widely applied in manufacturing metallic parts and components that require close clearances and smooth surface finishes. During this process, extreme heat is generated as well as very high cutting forces in the equipment and tooling. The application of lubrication and cutting fluids are needed to prevent the working pieces and tooling from burning, as well as to avoid undesirable residual tensile stresses, reduced fatigue strength, fissures, galling, and thermal distortions or inaccuracies, among others. A common machine operation for specific hard metals, such as titanium alloys, uses a flooding process with coolants and lubricants. This approach requires specialized equipment, large volumes of fluids, and high-power consumption pumps. When the machining lubricant is deposited to the grinding contact area, it will begin to experience nucleated boiling, which increases the heat transfer rate among the lubricant itself and the working piece; then, as the temperature is increased, a vapor film starts to develop in the working component and the lubricant interaction, acting as a thermal insulator preventing the heat transfer to the lubricant [15–17]. This results in a quick rise of the working piece temperature, subsequently burning the material's surface.

Currently, worldwide applications of harmful materials have been diminished or eliminated from manufacturing processes. Hence, researchers and industries have been working together to propose environmental alternatives or use of small amounts of fluids and lubricants. It is important to mention that some of those materials have a negative effect on operator health, such as skin or respiratory diseases.

The minimum quantity lubrication (MQL) technique is a good alternative to the known flood cooling technique, which has proved to be a promising methodology to improve surface quality, being cost-effective, reducing consumption of material, and having an impact on the environment, as well [18–20]. In this technique, the compressed air assisted minute droplets are applied by a nozzle, spraying on the cutting zone to provide cooling and lubrication effects [19]. The MQL system atomizer shoots the machining fluid or lubricant by applying high pressure air, and the lubrication particles are spayed on the working surface [21]. MQL offers comparable effects and results to those by using flood cooling if the cooling lubricant in MQL does not evaporate due to the heat generated in the grinding

process. The quantity of lubricant, delivery pressure, and working temperature are crucial factors when using MQL nanofluids [22,23]. A high working-cutting temperature leads to diverse tool wear mechanisms such as abrasion, high plastic deformation, attrition, and chipping, among others. The tool wear is minimized under other lubrication circumstances, particularly under MQL environment. Smaller average flank wear is recorded under pure MQL and nano-MQL (NMQL) working conditions.

Solid lubricants have demonstrated satisfactory attributes in grinding due to their high temperature withstanding capacity, non-toxicity, ease of application, and cost-effective characteristics. Despite all these good characteristics that solid lubricants possess, there is still a high necessity for flushing action and cleaning of tooling that make them less attractive than conventional liquid lubrication methodologies for certain applications. A proper machining fluid must be identified for the grinding process that will meet the primary functional requirements of a cutting fluid: lubrication, cooling, debris, chip removal, and providing a clean finish. Various studies have shown that $Al_2O_3$ is one of the best candidates for these experiments [5,24,25]. Using $Al_2O_3$ in its 20 nm scale and $\gamma$-phase (highest surface area) should most effectively reveal its nanostructures. Additionally, the high surface area will allow the usage of minimal fluid to reduce cost. Thermal conductivity is a vital attribute for heat removal from work-tool pieces. For instance, Zhang et al. [23] showed that MWCNT displayed the best work-tool piece heat removal results, while $ZrO_2$ and $MoS_2$ are more costly, less abundant, and have lower thermal conductivity than alumina. Additionally, reinforcing concentrations between 1 to 5 vol.% displayed variable results.

Khanafer et al. [24] studied the behavior of $Al_2O_3$ NMQL by developing simulations by using finite element analysis for Inconel alloy machining. According to their evaluations, incorporating $Al_2O_3$ nanostructures could reduce the average temperature of the working tool from 443 K (0 vol%) to 420 K (4 vol.%), which clearly demonstrated the high importance of incorporating nanostructures with MQL technique to reduce the cutting tool temperature. Studies by Sadeghi et al. [26] have shown that the most effective quantity of lubricant is 60 milliliters/h (19.8 drops/min) and a delivery pressure of 4 bars (58 psi). It was shown that synthetic oil performed better than natural lubricant in reducing grinding cutting forces and gave lower tangential forces. Furthermore, flow rates did not result in dispersion of mist, which contributes towards environmentally friendly manufacturing, reduced health hazards, and easier visibility of the working piece. Surface roughness in MQL grinding was greater due to the preserved sharpness of grits and less generated temperature. MQL grinding using synthetic lubricant generated a better quality on the surface (lower surface roughness without burned areas) and lower grinding forces than vegetable lubricant. Dongkum et al. [27] investigated the wheel tribological characteristics (wear) in dry, wet, and the MQL grinding process of casted iron. They observed that MQL reduced the grinding forces, improved the surface roughness, and prevented burning areas in the workpiece.

Investigations by Okokpujie et al. [28] on vegetable nanolubricants showed improvements in their rheological performance, resulting also in less cutting force required during the milling process. Similar research was conducted by Lim et al. [29] for hybrid nanolubricants for MQL in machining operations. There was graphene (G) and $Al_2O_3$ at 1.0 vol.% concentration at different nanostructure rations. It was observed that thermal conductivity improved 25% at a 20:80 (G—$Al_2O_3$) ratio, also showing a maximum enhancement of 45% in the coefficient of friction (COF) performance at a 60:40 ratio. According to Junankar et al. [22], MQL grinding using nanofluid had a better surface finish than just water and coolant. This is because the nanostructures reduced grinding forces and friction. Nanostructures tear off the bond material from the tool wheel, resulting in exposure of fresh active grits into the grinding zone.

This research aims to investigate the characteristics of creep feed grinding Ti-6Al-4V alloy with a green silicon carbide wheel, using a flood lubrication system with water soluble synthetic lubricant, MQL system with ester oil, and nano-MQL (NMQL) using a solution

of γ-alumina (20 nm) hydrophilic nanopowder homogeneously dispersed within ester lubricant, as well as finding the optimum parameters for creep feed grinding Ti-6Al-4V using the stated lubrication methods. These optimum grinding parameters should provide a high material removal rate with a clean, quality surface finish.

## 2. Material and Processes

### 2.1. Materials

The base metal to work with is a grade 5 titanium (Ti-6Al-4V) alloy. Alumina oxide ($Al_2O_3$) hydrophilic γ-phase from Sigma Aldrich (spherical, 20 nm in diameter, St. Louis, MO, USA) was used to prepare the working nanofluid. The γ-$Al_2O_3$ nanostructures are attractive economically and technologically because they possess low density, high hardness, excellent dimensional stability, and could be used for grinding and surface finishes. The γ-phase property increases the surface area reactivity of the nanostructures to a greater extent than α and β phase nanostructures. The hydrophilic nature of these nanostructures allows the fluid to coat the tool-work area properly due to its lower contact angle of 90 degrees. The nanolubricant in this study was prepared by dispersing the γ-$Al_2O_3$ hydrophilic nanostructures within the biodegradable ester oil (density 0.93 @ 25 °C) (Unist Inc., Grand Rapids, MI, USA), followed by extensive water bath ultrasonication (8 h), maintaining a constant temperature of 40 °C to prevent nanostructures' agglomeration and rapid sedimentation. The filler fraction used was 4.0 vol.%., which was chosen from previous experiments in our research group.

### 2.2. Grinding Operation

This study focuses on the application of nanofluid on the grinding wheel. A grinding wheel has speeds of 3000 rpm. Research shows that due to high speeds, an air pocket is created between the working piece and the tooling. These air pockets do not allow MQL to penetrate the working area. The current study focuses on creep feed grinding (Figure 1). In grinding operation, one of the main differences between titanium alloys and other metals is the endeavor of titanium at higher temperatures. Titanium can chemically react to the wheel material at a specific localized contact point of the wheel. Crucial factors to consider to prevent this are effective coolant, correct table feed rates so that the wheel is not on the surface for long durations, and the use of a green silicon grinding wheel [30–32]. Detailed information about the evaluation process is explained in the experimental section.

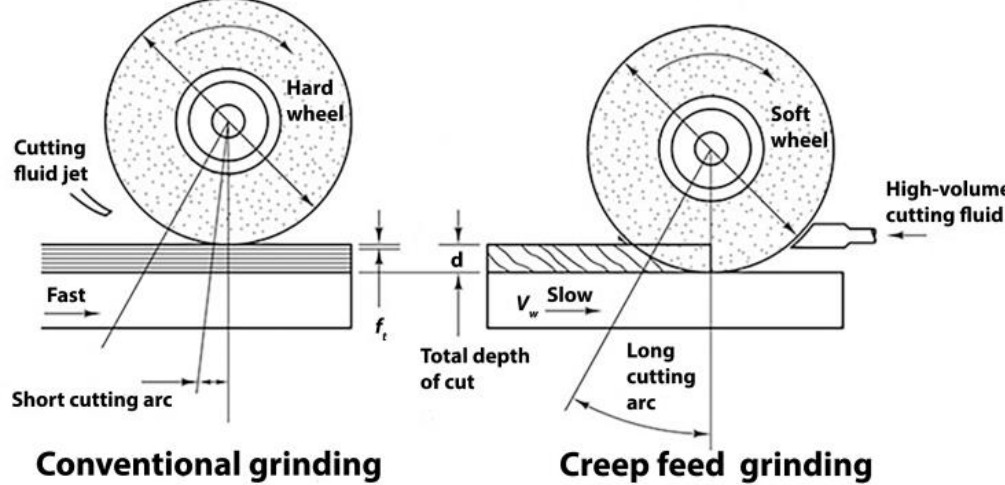

**Figure 1.** Graphical representation of conventional grinding vs. creep feed grinding.

## 3. Experimentation

Seeking an environmentally friendly nano-reinforced cutting fluid is a prime factor to address in investigations regarding titanium processes. The choice of an ester lubricant base $Al_2O_3$ nanofluid will meet this demand. The selection of a $\gamma$-phase small hydrophilic (>90-degree contact angle) $Al_2O_3$ nanostructure will allow the reduction of the amount of fluid needed to provide a good quality surface finish. Using a green silicon carbide wheel with a coarse 60 grit size will allow for the removal of titanium material. The MQL machine will allow the nanofluid to be evenly distributed on the tool-work piece at a set pressure and drop rate. Different grinding infeed and feed rates will be analyzed to produce optimum results.

The experiments are conducted using a hydraulic surface grinder that is used by small industries and craftsmen (Figure 2). The behavior of hydraulic surface grinders is analogous to that of high-end industrial grinders; therefore, it will also provide a clear understanding of the parameters to creep feed grind Ti-6Al-4V at an industrial level. The study includes the thorough understanding of the capabilities of the hydraulic surface grinder to creep feed grind Ti-6Al-4V. The operational parameters of the MQL system were also studied and their effects weighed. The MQL machine was used to study the effects of creep feed grinding Ti-6Al-4V with an ester oil base nanofluid. Creep feed grinding Ti-6Al-4V using a flood lubrication machine was used to have comparable results with the MQL methods. A synthetic oil was used as the grinding coolant. The operational parameters of the flood lubrication system were also studied, and their effects weighed.

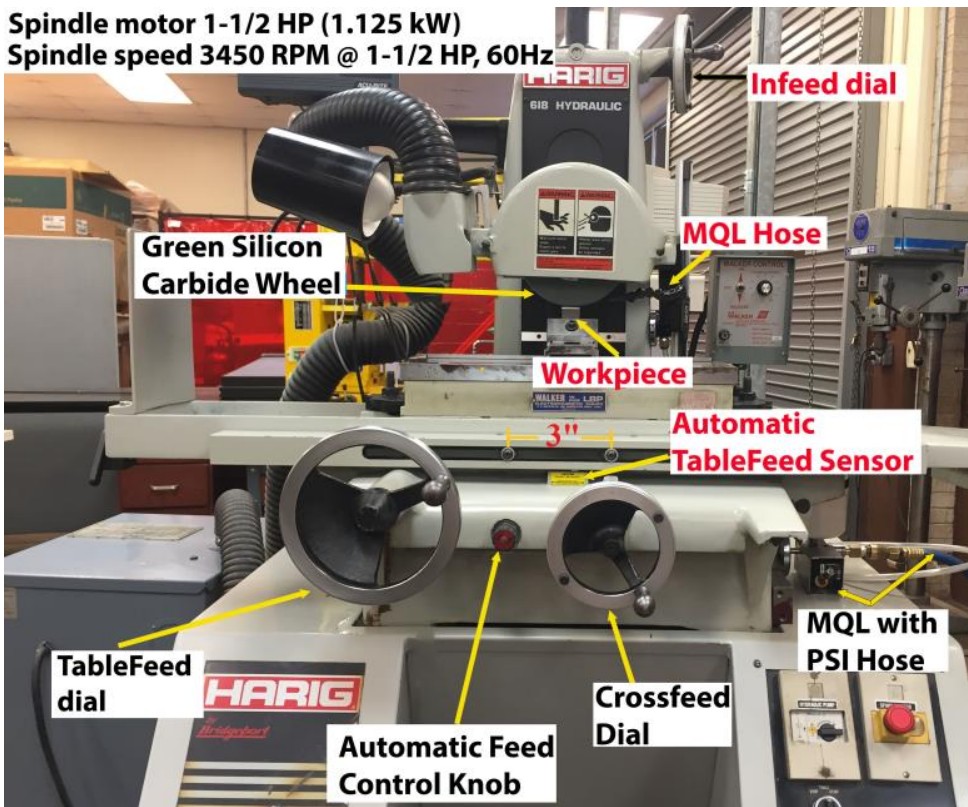

**Figure 2.** Hydraulic surface grinder characteristics with MQL setup.

### 3.1. Hydraulic Surface Grinder

The evaluations started by studying the capacity of the hydraulic surface grinder and the ideal parameters for the trials. The grinding machine is a Harig 618 Hydraulic Surface Grinder (Figure 2) with a spindle motor of 1.5 HP and spindle speed of 3450 rpm. Using the surface speed formula $V = \pi (D)(N)$, one may determine the surface speed of

the grinding wheel. The wheel diameter (D) is 0.1778 m. Therefore, the surface speed of the wheel is at 1926 m/min. The green silicon grinding wheel has a thickness of 12.7 mm, and only the edge of a grinding wheel does the cutting. Therefore, a 0.254 mm cross feed is chosen, which is the wheel's horizontal feed. Creep feed infeeds are high, though considering a spindle motor of 1.5 HP, the highest infeed possible without having the motor overheating is 1.905 mm. The machine has an automatic table feed rate with ranges between 1.219 m/min and 21.336 m/min. Since creep feed grinding requires low table feed rates, the preliminary tests were focused on finding the optimum table feed rate. In a preliminary stage of our investigation, we found the optimum feed rate zone is between 5.5 m/min and 6.7 m/min. Feed rates below 5.5 m/min or above 6.7 m/min resulted in undesirable characteristics such as excessive wheel wear, rough surface finishes, burn marks on the work piece, high temperatures, and overburden on the grinding wheel spindle motor.

### 3.2. Minimum Quantity Lubrication (MQL)

The minimum quantity lubrication machine used is Unist's MQL system, which disperses droplets of lubricant using pulses/min and pressure. The pulses are provided by an internal actuator that is activated through pressure. The pressure is non-electrical and is provided through an external pressurized tank. In our preliminary evaluations, we found desirable results at 200 pulses/min with a pressure of 414.685 kPa (60 PSI). As with the flood lubrication nozzle, our investigation found that having the nozzle at 38 mm from the tool-work interface and a nozzle angle of 4 degrees from the tool-work interface gave good performance machining (Figure 3).

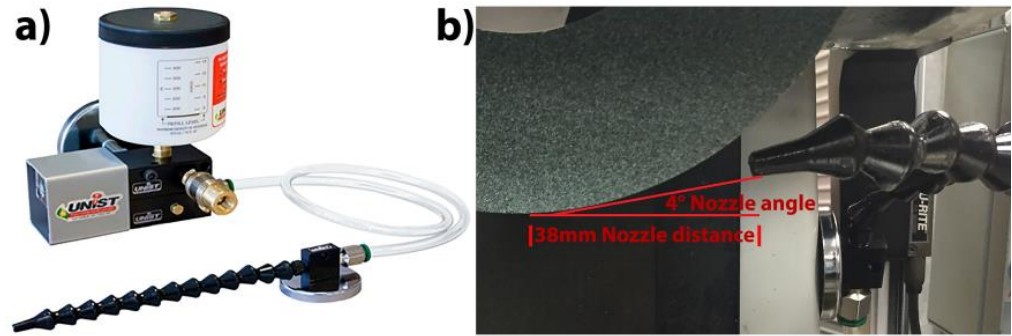

**Figure 3.** (**a**) Unist's Minimum Quantity Lubrication device system, (**b**) MQL nozzle with optimum distance and angle displayed.

### 3.3. Minimum Quantity Lubrication with Nanofluid

The nano-MQL (NMQL) design of experiments involve working with the developed $\gamma$-$Al_2O_3$ nanofluid. Selected parameters were based on our group research and previous research by Mahbubul et al. [33]. These parameters created a desirable nanofluid solution with properties that include higher thermal conductivity, lower viscosity, decrease in cluster size, and a higher density. The nanoparticle will penetrate the pores of the grinding wheel, creating a layer of nanostructures that have hydrophilic properties of contact angles below 90 degrees. This allows for the particle droplets to spread and attract themselves to the tool-work piece surfaces, creating a larger lubricated surface area (Figure 4).

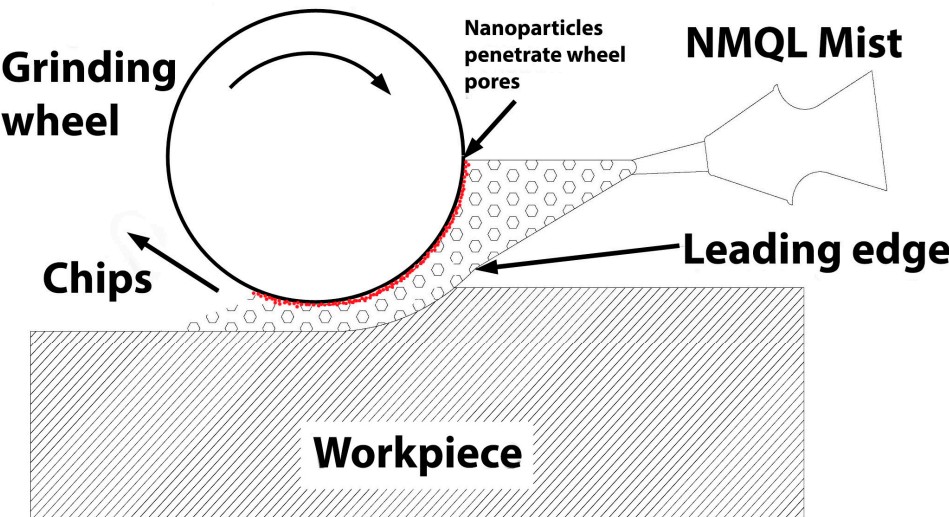

**Figure 4.** Schematic of nanostructures—wheel penetration while grinding operation is performed.

*3.4. Flood Pressurized Lubrication*

The hydraulic surface grinder (Figure 5a) worked with a flood lubrication machine shown in Figure 5b (Bijur Delimon's FluidFlex Pressurized Flood System). It uses a water-soluble coolant containing 97% water and 3% PowerChip 2000 coolant. The synthetic grinding coolant contains the following chemicals: polyalkylene glycol, triethanolamine, boric acid, and monoethanolamine. The system has an integrated electrical motor that requires electricity and provides pressure. Upon preliminary tests, we found that a pressure of 414.685 kPa (60 PSI) provided desirable results and avoided excess coolant usage. Nozzle distance and angle were important to consider for lubrication tool-work piece penetration. An optimum distance of 38 mm from the tool-work interface and a nozzle angle of 4 degrees from the tool-work interface was used in the experiments.

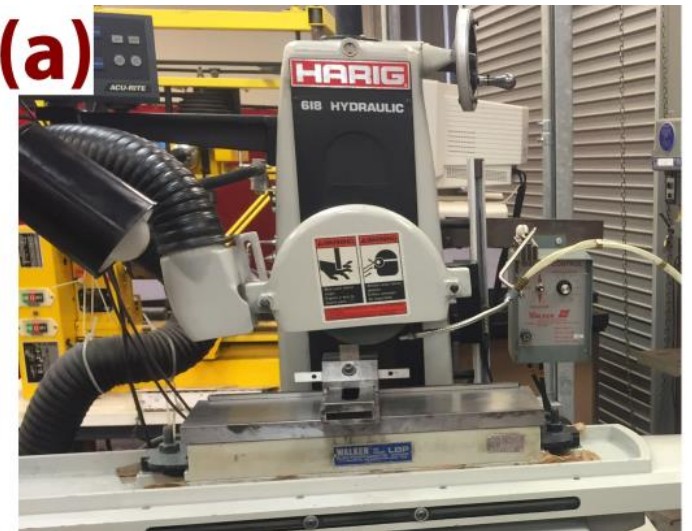
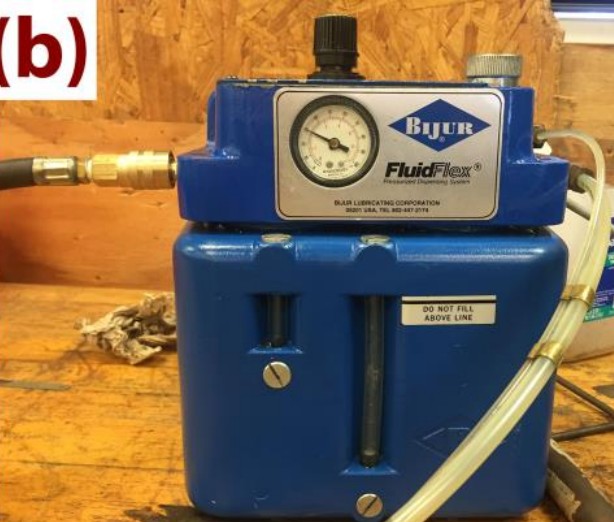

**Figure 5.** (**a**) Harig hydraulic surface grinder with flood lubrication system setup (**b**) Bijur Delimon's FluidFlex pressurized flood system.

*3.5. Preliminary Experiments*

The capacity of the hydraulic surface grinder and the most convenient parameters for the evaluations were assessed. The grinding machine has a spindle motor of 1.5 HP and spindle speed of 3450 rpm. Using the surface speed formula $V = \pi(D)(N)$, the surface

speed of the grinding wheel is determined. The wheel diameter (D) is 0.1778 m, spindle speed 3450 rpm, yielding a surface speed of 1926 m/min for the wheel. The green silicon grinding wheel has a thickness of 12.7 mm, and only the edge of a grinding wheel is used for the cutting. Therefore, 0.254 mm cross feed is chosen as the wheel's horizontal feed. Creep feed infeeds are high, though considering a spindle motor of 1.5 HP, the highest infeed possible without having the motor overheating is 1.905 mm. The machine has an automatic table feed rate with ranges between 1.219 m/min through 21.336 m/min. The preliminary investigation found an optimum feed rate zone, which is between 5.5 m/min and 6.7 m/min. Feed rates below 5.5 m/min or above 6.7 m/min resulted in undesirable characteristics such as excessive wheel wear, rough surface finishes, burn marks on the work piece, high temperatures, and overburden on the grinding wheel spindle motor.

The design of the experiment's parameters was obtained based on the Taguchi principle. A constant cross feed of 0.254 mm and three different infeeds values, with three different table feed rates for each infeed, were used. Each of these infeeds includes the three table feed rates for the MQL, NMQL, and the flood lubrication evaluations, which will result in a total of 27 Ti-6Al-4V samples. Evaluations were conducted with the grinding of Ti-6Al-4V at various infeeds and table feed rates. The machine has an automatic table feed rate with ranges between 1.219 m/min and 21.336 m/min. Since creep feed grinding requires low table feed rates, the preliminary tests were conducted to find the optimum table feed rate. Initial studies yielded the optimum feed rate zone between 5.5 m/min and 6.7 m/min. Feed rates below 5.5 m/min or above 6.7 m/min resulted in undesirable characteristics such as excessive wheel wear, rough surface finishes, burn marks on the work piece, high temperatures, and overburden on the grinding wheel spindle motor.

The three infeeds used in the experiments are 0.635 mm, 1.27 mm, and 1.905 mm. A total of three different table feed rates were tested for each infeed. These table feed rates are at 5.5 m/min, 6 m/min, and 6.7 m/min. Using these values, one may determine the material removal rate of grinding Ti-6Al-4V by using the material removal rate formula (Equation (1)).

$$Rmr = (V_w)\,(W)\,(d) \tag{1}$$

Here, $V_w$ is the table feed rate, W is the cross feed, d is the infeed, and multiplying these values gives the Rmr, which is the material removal rate in mm$^3$/min.

### 3.6. Flood Lubrication, MQL and NMQL Grinding Experiment

Table 1 shows the experimental setup parameters showing material removal rate calculations. The cross feed is constant at 0.254 mm. The tool-work interface was precisely recorded with a digital thermometer, and each sample was labeled and vacuum sealed.

The first stage consisted of testing nine samples using the FluidFlex flood lubrication machine with coolant consisting of one liter of water with 3% PowerChip 2000. The flood lubrication machine was set at 413.685 kPa (60 PSI). The second stage consisted of testing nine samples using an MQL device. The MQL pulse rate was set at 200 pulses/min and the MQL pressure was set at 413.685 kPa (60 PSI). There were 400 milliliters of ester oil lubrication consumed for all nine samples. The last stage of experiments involved evaluation of nine samples using the same MQL device, but in this case, we applied our nanofluid as lubricant. There were 400 milliliters of our pre-mixed ester oil base 4.0 vol.% nanofluid consumed for all nine samples.

**Table 1.** Experimental setup parameters showing material removal rate calculations.

| Experiment | Crossfeed (W) (mm) | Infeed (D) (mm) | Feed Rate ($V_w$) (mm/min) | Material Removal Rate (Rmr) (mm³/min) |
|---|---|---|---|---|
| 1 Flood | 0.254 | 0.635 | 5486.4 | 884.9 |
| 2 Flood | 0.254 | 0.635 | 6096.0 | 983.2 |
| 3 Flood | 0.254 | 0.635 | 6705.6 | 1081.5 |
| 4 Flood | 0.254 | 1.270 | 5486.4 | 1769.8 |
| 5 Flood | 0.254 | 1.270 | 6096.0 | 1966.4 |
| 6 Flood | 0.254 | 1.270 | 6705.6 | 2163.1 |
| 7 Flood | 0.254 | 1.905 | 5486.4 | 2654.7 |
| 8 Flood | 0.254 | 1.905 | 6096.0 | 2949.7 |
| 9 Flood | 0.254 | 1.905 | 6705.6 | 3244.6 |
| 10 NMQL | 0.254 | 0.635 | 5486.4 | 884.9 |
| 11 NMQL | 0.254 | 0.635 | 6096.0 | 983.2 |
| 12 NMQL | 0.254 | 0.635 | 6705.6 | 1081.5 |
| 13 NMQL | 0.254 | 1.270 | 5486.4 | 1769.8 |
| 14 NMQL | 0.254 | 1.270 | 6096.0 | 1966.4 |
| 15 NMQL | 0.254 | 1.270 | 6705.6 | 2163.1 |
| 16 NMQL | 0.254 | 1.905 | 5486.4 | 2654.7 |
| 17 NMQL | 0.254 | 1.905 | 6096.0 | 2949.7 |
| 18 NMQL | 0.254 | 1.905 | 6705.6 | 3244.6 |
| 19 MQL | 0.254 | 0.635 | 5486.4 | 884.9 |
| 20 MQL | 0.254 | 0.635 | 6096.0 | 983.2 |
| 21 MQL | 0.254 | 0.635 | 6705.6 | 1081.5 |
| 22 MQL | 0.254 | 1.270 | 5486.4 | 1769.8 |
| 23 MQL | 0.254 | 1.270 | 6096.0 | 1966.4 |
| 24 MQL | 0.254 | 1.270 | 6705.6 | 2163.1 |
| 25 MQL | 0.254 | 1.905 | 5486.4 | 2654.7 |
| 26 MQL | 0.254 | 1.905 | 6096.0 | 2949.7 |
| 27 MQL | 0.254 | 1.905 | 6705.6 | 3244.6 |

## 4. Results and Discussions

*4.1. Experimental Results for Flood Lubrication, MQL, and NMQL*

Table 2 represents the material removal rate, surface roughness across (Ra), surface roughness across (Rz), surface roughness along (Ra), surface roughness along (Rz), and grinding interface temperature for the evaluation performed. Using this data, line graphs were created to have a clearer understanding of the performance of the different grinding parameters using MQL, NMQL, and flood lubrications.

**Table 2.** Material removal rate and surface roughness across and along the working surfaces.

| Experiment | Material Removal Rate (Rmr) (mm³/min) | Surface Roughness across (Ra) (µm) | Surface Roughness across (Rz) (µm) | Surface Roughness along (Ra) (µm) | Surface Roughness along (Rz) (µm) | Grinding Interface Temp. (°C) |
|---|---|---|---|---|---|---|
| 1 Flood | 884.9 | 0.283 | 2.28 | 0.02060 | 0.133 | 34 |
| 2 Flood | 983.2 | 0.349 | 2.97 | 0.02360 | 0.150 | 37 |
| 3 Flood | 1081.5 | 0.356 | 2.66 | 0.02730 | 0.169 | 38 |
| 4 Flood | 1769.8 | 0.382 | 3.15 | 0.02730 | 0.174 | 53 |
| 5 Flood | 1966.4 | 0.400 | 3.58 | 0.02830 | 0.187 | 58 |
| 6 Flood | 2163.1 | 0.478 | 5.11 | 0.03480 | 0.215 | 59 |
| 7 Flood | 2654.7 | 0.506 | 3.47 | 0.03420 | 0.223 | 58 |
| 8 Flood | 2949.7 | 0.589 | 4.13 | 0.03920 | 0.245 | 39 |
| 9 Flood | 3244.6 | 0.599 | 4.35 | 0.03670 | 0.260 | 74 |

**Table 2.** *Cont.*

| Experiment | Material Removal Rate (Rmr) (mm³/min) | Surface Roughness across (Ra) (µm) | Surface Roughness across (Rz) (µm) | Surface Roughness along (Ra) (µm) | Surface Roughness along (Rz) (µm) | Grinding Interface Temp. (°C) |
|---|---|---|---|---|---|---|
| 10 NMQL | 884.9 | 0.276 | 2.11 | 0.02490 | 0.167 | 64 |
| 11 NMQL | 983.2 | 0.416 | 3.47 | 0.02860 | 0.189 | 67 |
| 12 NMQL | 1081.5 | 0.481 | 3.46 | 0.02920 | 0.193 | 69 |
| 13 NMQL | 1769.8 | 0.505 | 4.09 | 0.03370 | 0.210 | 89 |
| 14 NMQL | 1966.4 | 0.540 | 4.11 | 0.03470 | 0.212 | 93 |
| 15 NMQL | 2163.1 | 0.550 | 3.41 | 0.03450 | 0.213 | 95 |
| 16 NMQL | 2654.7 | 0.545 | 4.35 | 0.03820 | 0.228 | 110 |
| 17 NMQL | 2949.7 | 0.572 | 4.07 | 0.04320 | 0.292 | 118 |
| 18 NMQL | 3244.6 | 0.616 | 3.97 | 0.04760 | 0.280 | 121 |
| 19 MQL | 884.9 | 0.456 | 3.02 | 0.03280 | 0.188 | 71 |
| 20 MQL | 983.2 | 0.452 | 3.26 | 0.03220 | 0.206 | 74 |
| 21 MQL | 1081.5 | 0.515 | 4.29 | 0.03240 | 0.230 | 86 |
| 22 MQL | 1769.8 | 0.546 | 3.94 | 0.03410 | 0.229 | 97 |
| 23 MQL | 1966.4 | 0.553 | 3.96 | 0.03490 | 0.222 | 99 |
| 24 MQL | 2163.1 | 0.592 | 4.66 | 0.04180 | 0.250 | 100 |
| 25 MQL | 2654.7 | 0.873 | 6.09 | 0.04710 | 0.297 | 153 |
| 26 MQL | 2949.7 | 1.060 | 6.77 | 0.05680 | 0.326 | 159 |
| 27 MQL | 3244.6 | 1.490 | 8.74 | 0.07000 | 0.391 | 160 |

### 4.2. Grinding Interface Temperature Results

The tool-work interface temperature was measured during evaluations using an extech digital thermometer. Figure 6 depicts the temperature results for each sample test for the three methods of lubrication. Due to the internal pressurized motor, flood cooling had a substantial reduction in tool-work interface temperature over MQL and NMQL.

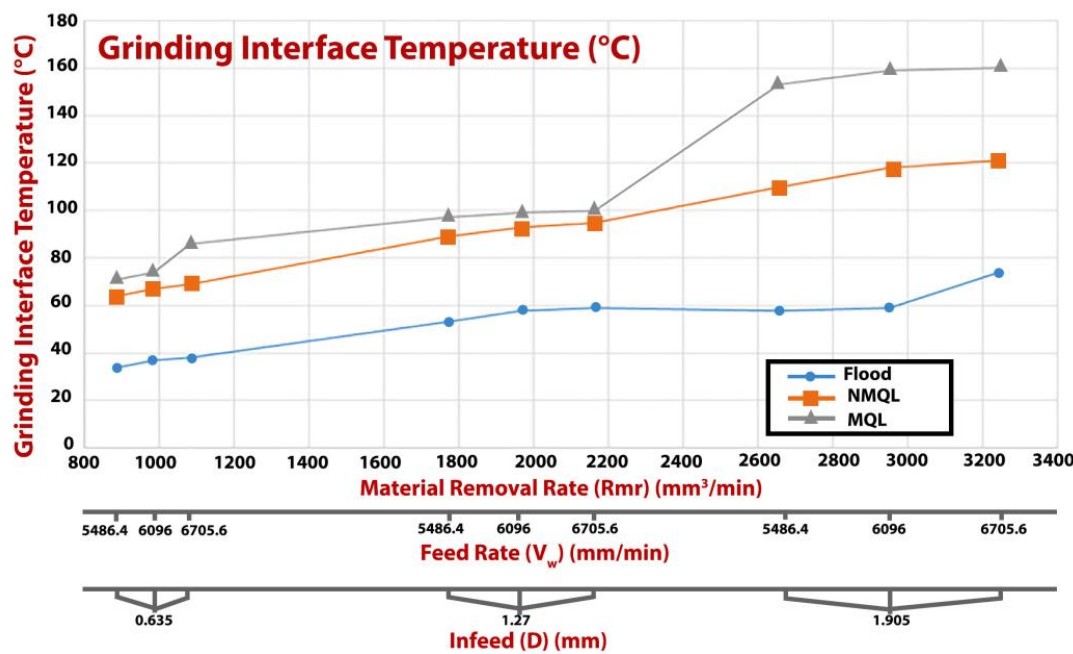

**Figure 6.** Grinding interface temperature of Ti-6Al-4V using flood, MQL, and NMQL.

### 4.3. Surface Roughness (Ra) across of Ti-6Al-4V Samples

The surface roughness measures the peaks and valleys created by the grinding wheel to determine the quality of the surface. Ra, which is the arithmetic average roughness, reflects the average height of roughness component irregularities from a mean line. The

across value is perpendicular to the cut. MQL, NMQL, and flood displayed similar behavior at infeeds of 0.635 mm and 1.27 mm (Figure 7). Nevertheless, at an infeed of 1.905 mm, the flood and NMQL performed almost identically, while MQL began to degrade in surface quality.

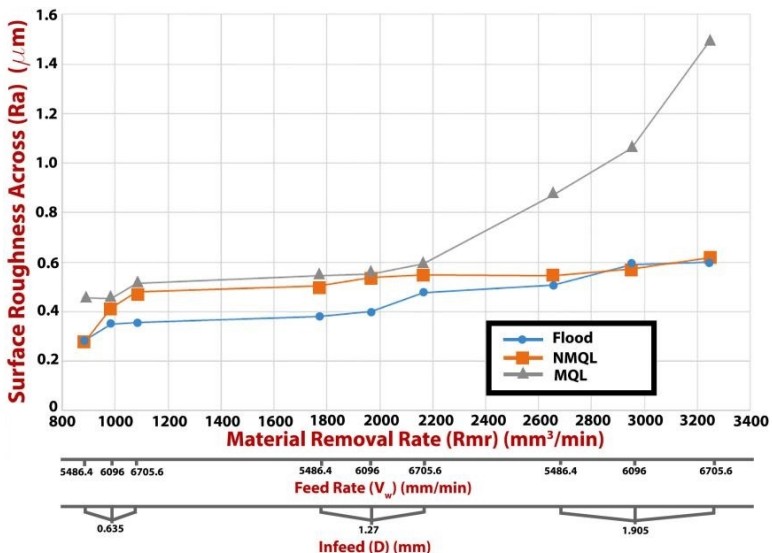

**Figure 7.** Surface roughness across (Ra) of Ti-6Al-4V samples using flood, NMQL, and MQL.

### 4.4. Surface Roughness (Rz) across of Ti-6Al-4V Samples

The Rz is known as the mean rough depth, which is the average distance between the highest peak and the deepest valley. The across value is perpendicular to the cut. MQL, NMQL, and flood showed similar performances at infeeds of 0.635 mm and 1.27 mm (Figure 8). Here, at an infeed of 1.905 mm, the flood and NMQL performed similarly; meanwhile, the MQL started to degrade in surface quality.

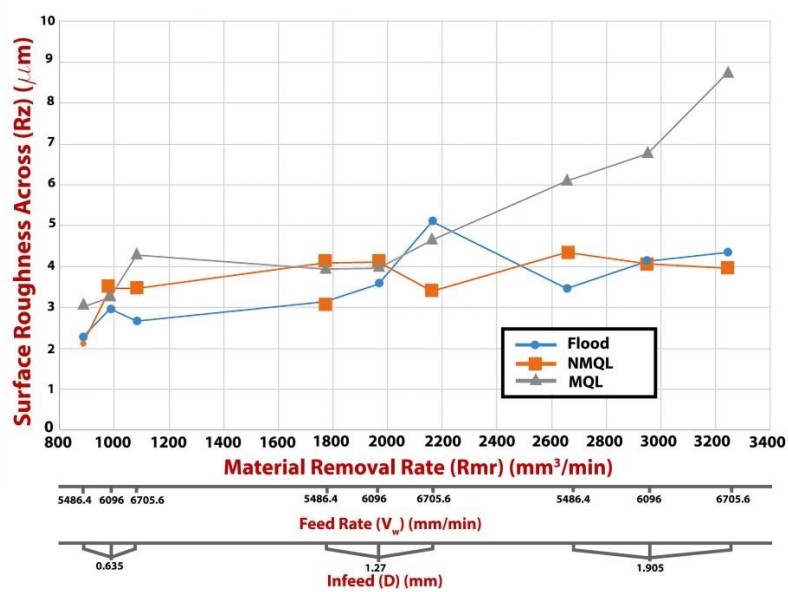

**Figure 8.** Surface roughness across (Rz) of Ti-6Al-4V samples using flood, NMQL, and MQL. The crossfeed (W) (mm) is a constant on all tests at 0.254 mm.

### 4.5. Surface Roughness (Ra) along of Ti-6Al-4V Samples

Surface roughness along of *Ti-6Al-4V sample* working pieces tested in the machining, using three lubrication configurations, MQL, NMQL, and flood, displayed the same

performance at infeeds of 0.635 mm and 1.27 mm (Figure 9). At an infeed of 1.905 mm, the flood and NMQL performed similarly, while the MQL started degrading the working surface quality.

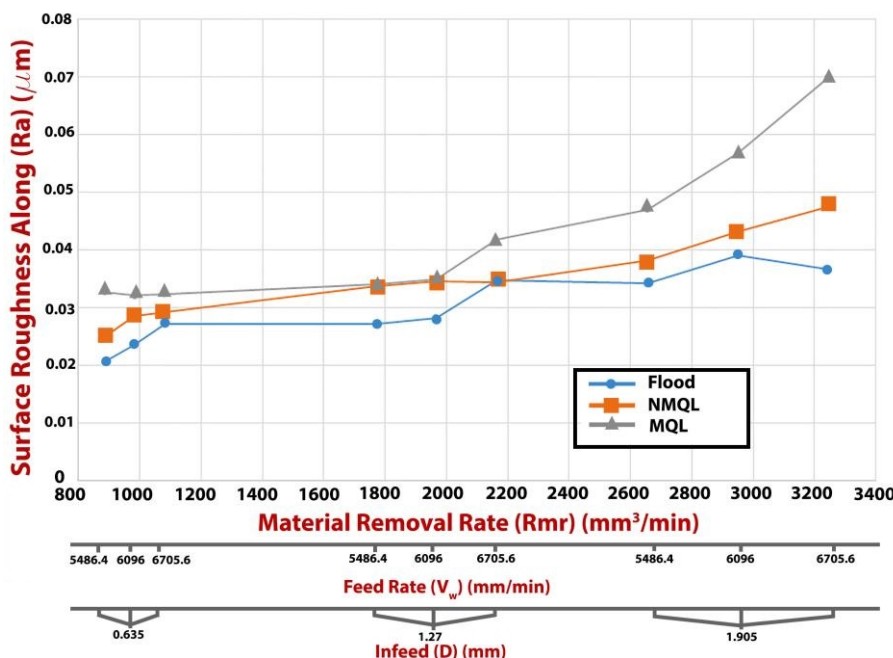

**Figure 9.** Surface roughness along (Ra) of Ti-6Al-4V samples using flood, NMQL, and MQL.

*4.6. Surface Roughness (Rz) along of Ti-6Al-4V Samples*

Surface roughness along of *Ti-6Al-4V sample* working pieces tested in the machining, using three lubrication configurations, MQL, NMQL, and flood, displayed the same performance at infeeds of 0.635 mm and 1.27 mm (Figure 10). At an infeed of 1.905 mm, the flood and NMQL performed similarly, but the MQL started degrading the working surface quality.

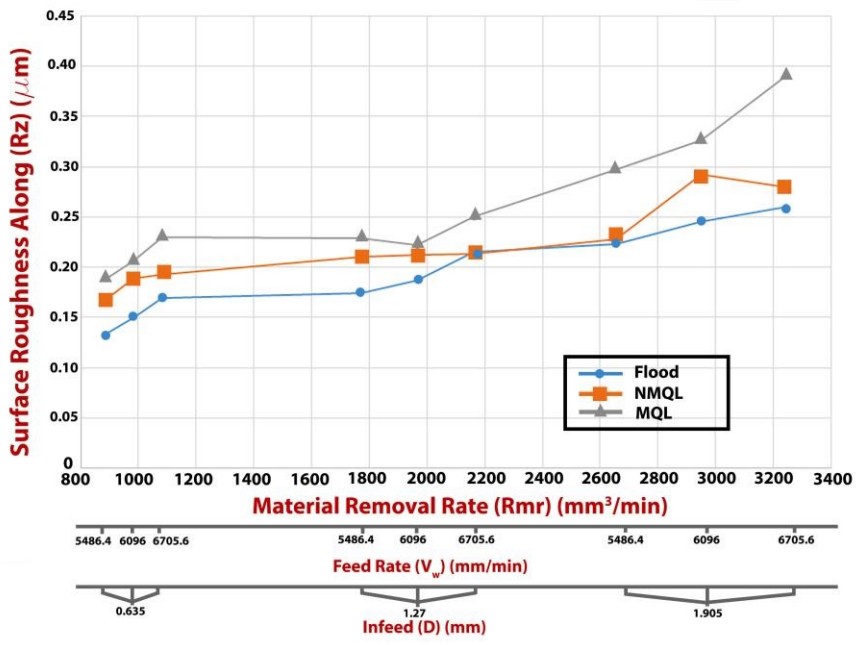

**Figure 10.** Surface roughness along (Rz) of Ti-6Al-4V samples using flood, NMQL, and MQL. The crossfeed (W) (mm) is constant on all tests at 0.254 mm.

### 4.7. Lubrication Methods Wheel Wear

A comparative study of grinding wheel wear for different lubrication methods was conducted to find the lubrication methods that produce minimum wear in infeed grinding. Figure 11 represents the grinding wheel wear after the material removal of each sample at the corresponding grinding parameters.

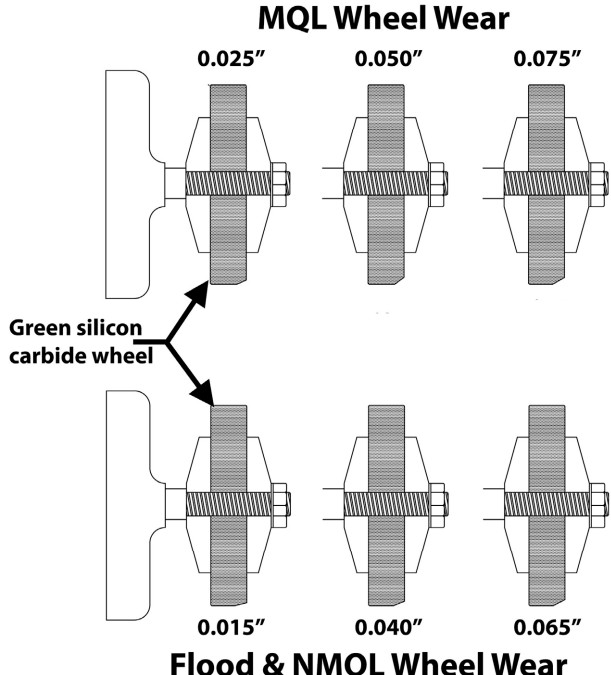

**Figure 11.** The wheel wear of the MQL test (**top**) and the wheel wear of the flood and NMQL tests (**bottom**).

### 4.8. Comparative Microscopic Results

Figure 12 represents a microscopic topographic view at 20× magnification. Figure 12a is sand polished and Figure 12b is a rough band saw cut, and Figure 13 shows a visual collage of nine samples with the highest table feed rates with the corresponding infeeds with the three lubrication methods.

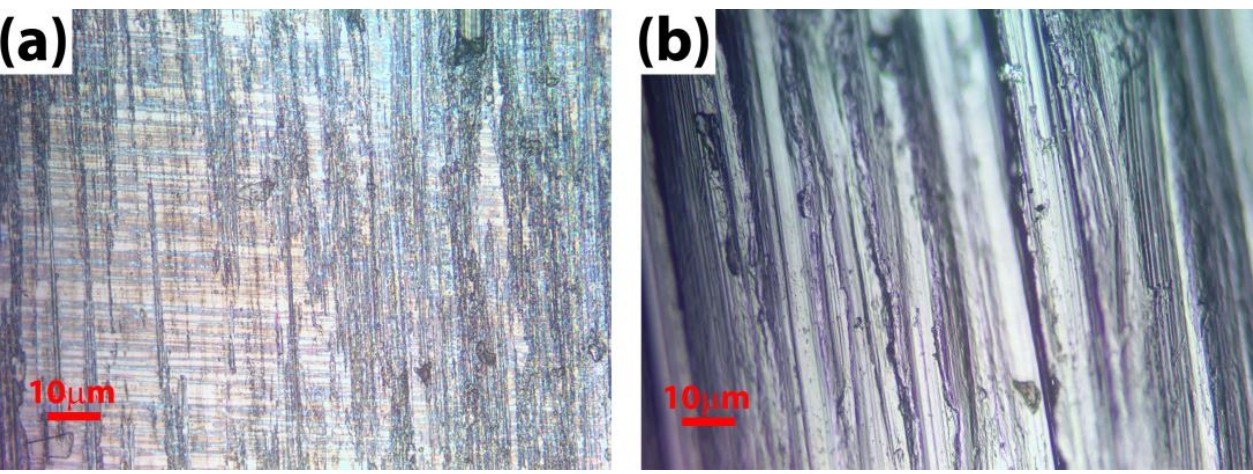

**Figure 12.** (**a**) Sand polished microscopic image at 20×, (**b**) Band saw cut microscopic image at 20×.

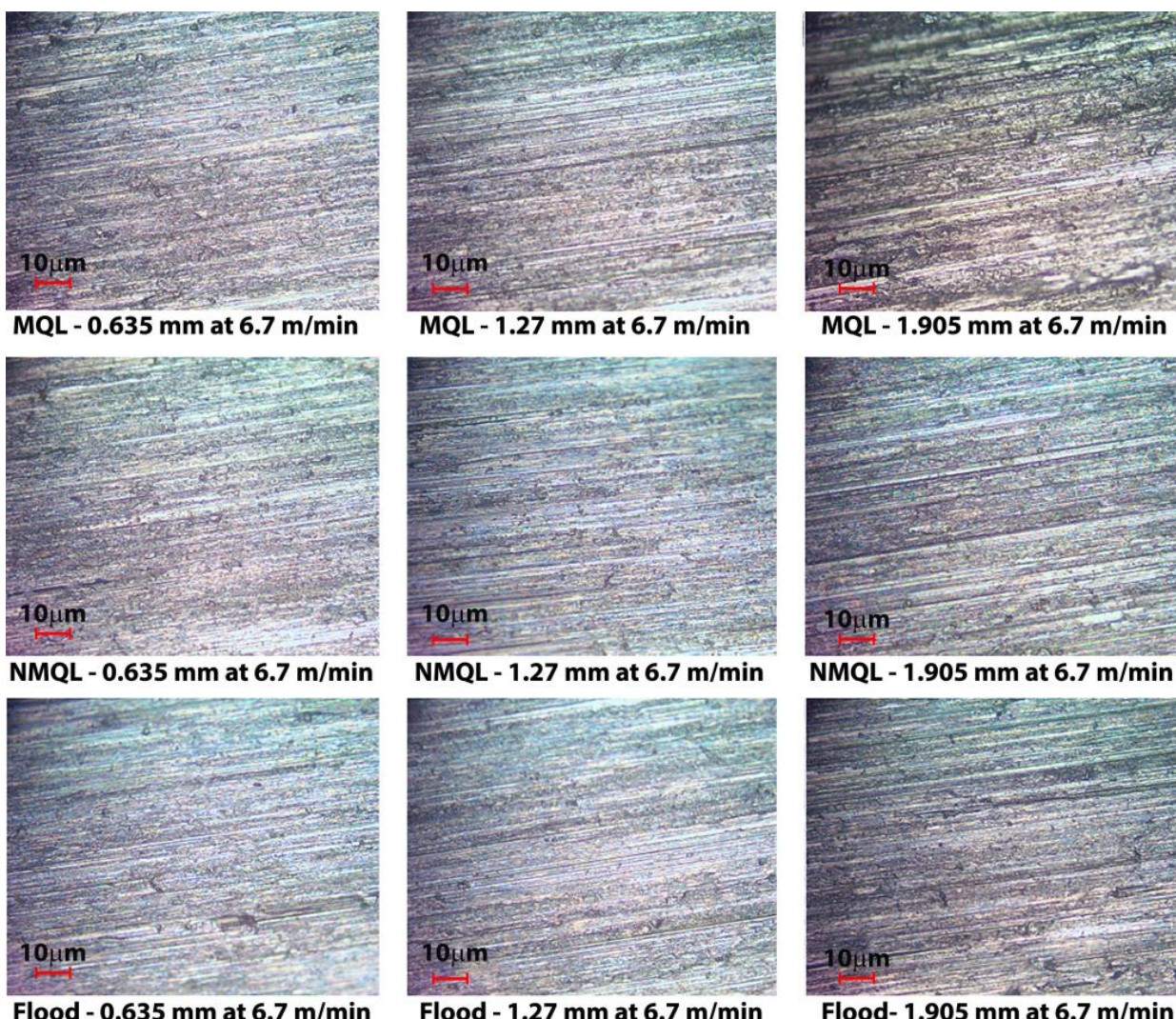

**Figure 13.** Microscopic images of surface quality produced in the machining by the MQL, NMQL and Flood lubrication techniques.

Figure 13 shows the cutting configurations at various infeeds. Results indicate that the NMQL and flood-lubricated machining produce similar quality in surface finishing and are superior to MQL-lubricated samples.

## 5. Conclusions

The present research shows the optimum parameters for creep feed grinding of Ti-6Al-4V alloy with a hydraulic surface grinder. Additionally, the research provided an understanding of the nature of grinding Ti-6Al-4V alloy using a MQL machine with ester oil, a NMQL with $\gamma$-Al$_2$O$_3$ nanofluid, and using a flood lubrication with a water-soluble synthetic coolant. From evaluations, it was concluded that at 0.635 mm and 1.27 mm infeeds, the three lubrication methods performed similarly. At an infeed of 1.905 mm, MQL did not provide desirable quality, though NMQL and flood lubrication performed practically the same. MQL is preferred over flood lubrication due to the environmental hazards of using water soluble synthetic coolants. Furthermore, MQL does not require any source of electricity to operate, which is a positive advantage over flood lubrication. An MQL system requires 60% less coolant than flood lubrication, having a direct impact in cost. As a comparison, to creep feed grind nine titanium specimens, the MQL system required 400 milliliters of biodegradable ester oil, while flood lubrication required 1000 milliliters of a toxic water-soluble synthetic oil.

The MQL's operating parameters involve having the pulse rate set at 200 drops/min and 413.685 kPa (60 PSI) of pressure. The nozzle should be set 38 mm away from the tool-work interface at a nozzle angle of 4 degrees from the base of the tool-work interface. With MQL using ester oil at an infeed of 0.635 mm and a table feed rate of 6.7 m/min, a surface roughness across (Ra) of 0.515 μm was achieved. At an infeed of 1.27 mm and a table feed rate of 6.7 m/min, a surface roughness across (Ra) of 0.592 μm was achieved. At an infeed of 1.905 mm and a table feed rate of 6.7 m/min, a surface roughness across (Ra) of 1.490 μm was obtained.

With the NMQL technique, with an infeed of 0.635mm and a table feed rate of 6.7 m/min, a surface roughness across (Ra) of 0.481 μm was achieved. At an infeed of 1.27 mm and a table feed rate of 6.7 m/min, a surface roughness across (Ra) of 0.550 μm was achieved. At an infeed of 1.905 mm and a table feed rate of 6.7 m/min, a surface roughness across (Ra) of 0.616 μm was achieved. With flood lubrication, at an infeed of 0.635 mm and a table feed rate of 6.7 m/min, a surface roughness across (Ra) of 0.356 μm was achieved. At an infeed of 1.27 mm and a table feed rate of 6.7 m/min, a surface roughness across (Ra) of 0.478 μm was achieved. At an infeed of 1.905 mm and a table feed rate of 6.7 m/min, a surface roughness across (Ra) of 0.599 μm was achieved.

The lowest surface roughness (Ra) is produced due to reduction in friction and promoted ease of material chips and debris sliding over the tool surface. Regarding surface roughness, it was also found that NMQL was effective for reducing surface roughness values of ground workpieces. The size of nanostructures was a more critical parameter to influence surface roughness than the volumetric concentration, and smaller nanoparticles could be more effective for producing a smoother surface in the case of NMQL. The addition of $\gamma$-$Al_2O_3$ to base lubricant improved the wear resistance and decreased the friction among tooling and working pieces. This may be attributed to the lubricant nature of $Al_2O_3$ nanostructures.

The results proved conclusively that MQL is preferred in creep feed grinding Ti-6Al-4V below an infeed of 1.905 mm. This conclusion is based on environmental, economic, and qualitative results. Our team established the optimum grinding parameters for removing Ti-6Al-4V with a hydraulic surface grinder. These parameters provide the quickest material removal rate while still maintaining industrial quality.

**Author Contributions:** Conceptualization, J.J.T.-T. and I.A.E.; methodology, J.J.T.-T. and I.A.E.; validation, J.J.T.-T. and I.A.E.; formal analysis, J.J.T.-T. and I.A.E.; investigation, J.J.T.-T. and I.A.E.; resources, J.J.T.-T. and I.A.E.; data curation, J.J.T.-T. and I.A.E.; writing—original draft preparation, J.J.T.-T. and I.A.E.; writing—review and editing, J.J.T.-T. and I.A.E.; visualization, J.J.T.-T.; supervision, J.J.T.-T. and I.A.E.; project administration, J.J.T.-T. and I.A.E. All authors have read and agreed to the published version of the manuscript.

**Funding:** This research received no external funding.

**Acknowledgments:** Authors acknowledge the support from UTRGV Engineering Technology Department and from Estuardo de los Reyes and Jesus Salinas.

**Conflicts of Interest:** The authors declare no conflict of interest.

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
