# Peer review of "Comparative Cutting Fluid Study on Optimum Grinding Parameters of Ti-6Al-4V Alloy Using Flood, Minimum Quantity Lubrication (MQL), and Nanofluid MQL (NMQL)"

_lubricants, doi:10.3390/lubricants11060250_

Round 1

Reviewer 1 Report

I have reviewed your work and can conclude that this is an interesting topic. The work has a good structure however there are few issues that are not explained clearly enough and some that need major corrections:

(1)Firstly, it is necessary to check the titles of the images and tables in the article to ensure that they meet the needs of the magazine.

(2)You need presented more detailed information about Ti-6Al-4V, chemical composition etc..

(3)The content of the section 2.2 should be reduced, some well-known content can be deleted, and some content that cites the research results of others should be included in the introduction. This section mainly introduces the content related to the experiment.

(4)The measurement of grinding temperature should be described in detail in the experimental section.

(5)Grinding force is very important, why is it not measured and analyzed in this article?

(6)The experimental data process in this article is detailed and rich, but the biggest regret is that the analysis of the lubrication mechanism of nanofluids in this article is insufficient. Therefore, the reviewers believe that some content should be added, and it is better to analyze the lubrication mechanism of different grinding media in the grinding process in combination with some necessary analysis and testing, that is, how do nanoparticles act on the grinding interface in the grinding interface to reduce the grinding temperature and improve the surface quality of the workpiece?

The English writing of this article meets the requirements.

Author Response

Thanks for your comments and recommendations.

Our response is in the attached file.

Regards,

Reviewer 2 Report

My comments are in the file attached.

Author Response

(The authors gave the same response as above.)

Round 2

Reviewer 2 Report

Accepted!

Author Response

Thanks for your time and attention for reviewing our manuscript.

We appreciate your approval decision.

Regards,